# Characterising the HIV self-testing market in Kenya: Awareness and usage, barriers and motivators to uptake, and propensity to pay

Rebecca L. West[1,2]*, Lily Freeman[1], Charlotte Pahe[3], Harmon Momanyi[3], Catherine Kidiga[4], Serah Malaba[4], Joanna Ciecielag[5], Mary-Clare Ridge[1], Emma Goldwin[1], Heather Awsumb[6], Sunny Sharma[1]

1 Ipsos, London, United Kingdom, 2 Department of Global Health, Boston University School of Public Health, Boston, Massachusetts, United States of America, 3 Population Services Kenya, Nairobi, Kenya, 4 Population Services International, Nairobi, Kenya, 5 Ipsos, Warsaw, Poland, 6 Population Services International, Washington, D.C., United States of America

* rebecca.west@ipsos.com

**Data Availability Statement:** All relevant data are within the paper and its Supporting Information files.

## Abstract

HIVST has a key role in ensuring countries meet their 95-95-95 goals. For HIVST to be sustainable, we should explore sharing costs with users as well as the overall experience. This research explores why a consumer would use HIVST and willingness to pay for HIVST through surveying 1,021 participants 18–35 living in Nairobi or Kisumu who were not diagnosed as HIV positive and who are not currently taking PrEP for HIV. A majority (89.8%) would pay 100 KSH and 64.7% would pay 300 KSH, at higher prices likelihood of paying dropped sharply. Price reduction or subsidization coupled with interventions to address the identified barriers may increase HIVST uptake. We identified 5 distinct groups based on willingness to pay and drivers/ barriers to HIVST uptake. These were created using dimension reduction, hierarchical clustering, and k-means analysis to group respondents. 79% of participants had ever heard of HIVST, and 24% had ever used HIVST. The 5 groups included active users, unlikely users, and three segments interested in HIVST with different barriers: need for HCP support, need for increased privacy/confidentiality, and fear of positive result/ disclosure.

## Introduction

Kenya has made great strides toward achieving the UNAIDS 95-95-95 targets to end the HIV epidemic by 2030 [1]. In 2018, 96% of individuals with knowledge of their HIV status were on ART and 90.6% of those on ART were virally suppressed; however, there is still progress to made on the first 95: 20% of people living with HIV aged 15–64 years did not know their HIV status in 2018 [2]. HIV self-testing (HIVST) has been demonstrated to increase testing uptake and close this gap [3, 4] by allowing people to test when, where and with whom they want to [5]. HIVST has been recommended by the WHO as an additional approach to HIV counselling and testing services since 2016 [6], with multiple prequalified HIVST kits on the market, and ongoing attempts at price reductions including through provision of subsidies.

**Funding:** Population Services International received funding for this research from the Children's Investment Fund Foundation, UNITAID, and the Elton John AIDS Foundation. The funders had no role in study design, data collection and analysis, decision to publish, or preparation of the manuscript.

**Competing interests:** The authors have declared that no competing interests exist.

Multiple countries across sub-Saharan Africa have included HIVST in their testing guidelines [7], such as Kenya, which officially launched HIVST in their public sector strategy in 2017 [8], and efforts have been made to establish a sustainable market for HIVST through the private health sector, community, and workplace settings. HIVST has shown to be safe, accurate and acceptable across sub-Saharan Africa [9–13]; and in Kenya across diverse user groups [14–17]. However, there is limited data available on uptake of HIVST across Kenya, and existing data suggest HIVST uptake in Kenya has been low: KENPHIA data from 2018 showed that 4.1% of adults who had ever tested for HIV had used HIVST [4], and more recent data on prevalence of HIVST use and characteristics of users is limited. To understand the potential to create a sustainable private market for HIVST, there is a need to better understand the current landscape of HIVST users in Kenya, what proportion of potential users are undecided, and what proportion are unlikely to ever use HIVST. For any undecided HIVST users, identifying their unique motivators and barriers to use HIVST can inform efforts to better tailor distribution methods and communications campaigns.

On behalf of Population Services International and Population Services Kenya, as part of the HIVST Challenge Fund project, commissioned Ipsos (a global social and market research organization) to carry out a survey among 1,021 participants 18–35 living in Nairobi or Kisumu counties who have not been diagnosed by a healthcare professional as HIV positive and who are not currently taking HIV pre-exposure prophylaxis. This research sought to characterise those in this audience in Kenya in several ways: awareness (the proportion who have heard of HIVST), current usage (proportion who had ever used HIVST, frequency of HIVST use), and consideration (the proportion who have heard of self-testing but have not yet used it, and why).

To investigate "consideration," we conducted an attitudinal segmentation to identify and quantify unique groups of potential HIVST users and tease apart their unique barriers and facilitators to testing. Segmentation is an analytic tool borrowed from social and commercial marketing which is increasingly being used to tackle public health challenges, including HIV [18, 19], voluntary male medical circumcision [20], family planning [21], and tuberculosis [22]. Segmentations can be used for developing targeted health information, programmes and messages for discrete sub-groups of a population that are similar in attitudes and behaviour [23–25], rather than relying on demographics (like age, gender) to develop public health interventions and communications. In the case of HIVST, a segmentation can be used to quantify groups within a population who are already using HIVST (e.g., early adopters), who are unlikely to ever use HIVST, and identify key drivers and barriers to use for potential users.

Lastly, we examined propensity to pay for HIVST: high prices in the retail sector may impede access for those who would most benefit from HIVST [26]; we therefore examined how price of kits may influence formation of a sustainable HIVST market, and how propensity to pay intersects with motivators/barriers to use HIVST and sociodemographic characteristics.

## Methods

### Survey design

The questionnaire was designed by Ipsos in close collaboration with Population Services Kenya and Population Services International and took an average of 30 minutes to administer to participants. Interviews were carried out in English and Kiswahili. The topics included in the questionnaire were sexual lifestyles, HIV knowledge and attitudes, knowledge of HIVST, usage of HIVST, motivations and barriers to use HIVST, and propensity to pay for HIVST at varying price points. Propensity to pay was asked using the Gabor-Granger method, whereby all participants were first asked whether they would pay for HIVST at a median price point

(500 KSH, equivalent to approximately 5 USD–using a 100:1 exchange rate) and then at a price increasing or decreasing by 200 KSH intervals based on whether they would pay at 500 KSH (from 100–900 KSH, equivalent to 1 and 9 USD, respectively).

## Sample

Eligibility criteria for this research included being aged 18–35 and living in either Nairobi or Kisumu. Those who were already on HIV pre-exposure prophylaxis or diagnosed by a health-care professional as HIV-positive were excluded from participation. A total of 1,021 participants were recruited to ensure a robust analytical framework with which to conduct the segmentation analysis. The sampling plan was developed by identifying the size and profile of this clearly defined target audience using the 2014 Kenya Demographic and Health Survey 2014. The profile of this population estimate was adjusted to remove both the estimated number of people within this age group living with HIV using prevalence estimates from KEN-PHIA [2] and the estimated number of people on HIV pre-exposure prophylaxis using PrEPWatch [27]. Quotas were set by age and gender. S1 Appendix outlines the target and achieved sample. The data is not weighted as the profile of the sample matched that of the target audience so that it is broadly representative of the population of this audience.

## Data collection

Interviews were carried out by Population Services Kenya in the 17 sub-counties of Nairobi and 2 sub-counties of Kisumu in which the Challenge Fund project was being implemented. Face-to-face interviews were conducted at the household level in participants' homes in February 2022 using computer assisted personal interviewing via an online data capture platform (SurveyCTO) on smartphones and tablets. Interviewers randomly selected households using the following approaches: first, a central location point (usually a landmark) was identified in each village visited. Using the left-hand rule approach, the first household was randomly selected using the date score (combining the two digits of the date, e.g., if research was carried out on the 12th, the sum of the two digits of the date was done (1+2 = 3) and the total sum 3 was taken as the random number for the first household). The interviewers would then use this random number to identify which house to start interviews (e.g., the 3rd house to the left of the starting point). Interviewers would ask if anyone from the target age (18–35) was present; if yes, they would start the screening process. After a successful interview, interviewers would skip 4 households and attempt an interview in the 5th house keeping the left-hand rule, and so forth. If there were no eligible participants in the household, interviewers would proceed to the next house until a successful interview was completed; and then subsequently apply the 5th household rule until the quota for that village was met or the day ended.

## Ethical approval

Verbal consent was obtained from all participants prior to participating in the survey. Ethics approval for this research was provided by the Ethics and Scientific Review Committee (ESRC) (P1105-2021) at Amref Health Africa, Nairobi, Kenya.

## Analysis

**Awareness and usage of HIVST.** Descriptive statistics were run to quantify awareness of HIVST (e.g., ever heard of HIVST) and usage (ever used HIVST, number of times using HIVST).

**Factors influencing consideration of HIVST: Segmentation analysis.** A segmentation analysis was conducted that included a series of steps described in detail in Johansson & Sheth [28]. First, key segmentation variables were identified to determine the constructs on which segments would be differentiated. The list of potential segmentation variables was then revised to exclude those which showed little differentiation across participants, had missing data, or were highly correlated with other variables under consideration. Factor analysis was performed to reduce the number of segmentation variables and to link inter-correlated statements together in 'joint' dimensions. Canonical correlation was employed to link motivators and barriers to use HIVST and combine them in aggregated measures. Based on canonical variables created, exploratory hierarchical clustering was used to identify the optimum number of segments and to develop initial cluster centres, which were then refined using k-means clustering to create possible segments [29]. Each developed segment solution was evaluated statistically (significance of canonical roots and amount of variance explained, 'goodness to fit' criteria based on segment distances, segment frequency, cross-country segment distribution, predictability). Several configurations of clusters were tested (with four and five segments), and the chosen solution was selected based on statistical features, as well as its ease of understanding and practical usability. The segments in the chosen solution were then profiled on the segmentation variables, and other demographic or attitudinal variables of interest. These analyses were performed using SAS v9.4 (SAS Institute Inc, 2013) and SPSS v24 (IBM Corp, 2016).

Differences between segments were tested using one-way ANOVA. When a difference at alpha level >.05 was detected in the ANOVA, differences between groups were tested using a Bonferroni correction (chosen due to the differing sample sizes of each segment) using Segment 1 as the reference group. Analyses were performed using RStudio v4.1 (RStudio Team, 2020).

**Propensity to pay.** Propensity to pay for HIVST across each price point was imputed for based on the highest or lowest price point at which participants would pay (for example, if a respondent would not pay 300 KSH, it was imputed that they would not pay at 100 KSH). A four-point scale was collapsed into two variables, "would buy" or "would not buy". Descriptive statistics were run to show at which prices participants would pay for HIVST from 100–900 Kenyan shillings (KSH) at 200 KSH intervals.

## Results

### Sample characteristics

A total of n = 1021 individuals were interviewed; 62 were screened out for various reasons: being under age 18 (n = 35), not agreeing to consent (n = 3), currently on ART or PrEP (n = 11) and being HIV-positive (n = 11). Of the final sample, 84.7% of participants were from Nairobi and 15.3% from Kisumu; 49.5% were male and 50.5% were female. Participants resided in urban (83.0%), peri-urban (14.1%) and rural (2.9%) areas. 43.0% of participants were aged 18–24, 34.6% were aged 25–30, and 22.4% were aged 31–35. The highest level of education obtained for most participants was either secondary school/A level (39.6%), college (32.2%) or university (16.5%) with those remaining attending primary school (11.2%) or never attending school at all (0.5%); for more details on sample characteristics reference Table 1.

### Awareness and usage of HIVST

78.6% of all participants had heard of HIVST before (Table 2). 25.2% had ever used a HIVST self-test kit and 15.6% all participants had ever purchased an HIVST kit before, either for themselves

**Table 1. Final sample characteristics.**

| Sample Characteristic | Total sample |
|---|---|
| | **n = 1,021** |
| **County** | |
| Nairobi | 865 (84.7%) |
| Kisumu | 156 (15.3%) |
| **Gender** | |
| Male | 505 (49.5%) |
| Female | 516 (50.5%) |
| **Age** | |
| 18–24 | 439 (43.0%) |
| 25–30 | 353 (34.6%) |
| 31–35 | 229 (22.4%) |
| **Setting** | |
| Urban | 847 (83.0%) |
| Peri-urban | 144 (14.1%) |
| Rural | 30 (2.9%) |
| **Main source of Income** | |
| Parent/relative support | 141 (13.8%) |
| Farming | 3 (0.0%) |
| Private sector | 130 (12.7%) |
| Civil service/government | 33 (3.2%) |
| Spousal support | 27 (2.6%) |
| Casual work | 146 (14.3%) |
| Domestic work | 26 (12.5%) |
| Informal sector | 413 (40.4%) |
| Other* | 32 (3.1%) |
| **Education Level** | |
| Never attended school | 6 (0.0%) |
| Primary | 114 (11.2%) |
| Secondary/A Level | 404 (39.6%) |
| College | 329 (32.2%) |
| University | 168 (16.5%) |
| **Money earned in a year (KSH)** | |
| Less than 10,000 | 183 (17.9%) |
| 10,001–30,000 | 164 (16.1%) |
| 30,001–50,000 | 105 (10.3%) |
| 50,001–70,000 | 109 (11.0%) |
| 70,001–90,000 | 115 (11.0%) |
| 90,001–110,000 | 115 (11.0%) |
| 110,001 and above | 229 (23.0%) |

or someone else. Of all participants, 21.1% had used HIVST 1–3 times, 1.9% had used HIVST 4–10 times, and >1% had used HIVST more than 10 times in the last 12 months. 32% of those who had ever heard of HIVST had used it (257/803). 61.8% of those who had used HIVST reported ever purchasing a test (159/257). Of HIVST users, 83.6% had used HIVST 1–3 times (215/257), 7.8% had used HIVST 4–10 times, and 2.3% had used HIVST more than 10 times in the last 12 months. HIVST users primarily accessed tests through pharmacies (119/257, 46.3%), clinics/hospitals (59/257, 22.9%) and sexual partner distribution (25/257, 9.7%).

**Table 2. HIVST awareness and usage.**

| Awareness and usage | Base: Total sample | Base: Ever used HIVST |
|---|---|---|
| | n = 1,021 | n = 257 |
| Aware of HIVST | 803 (78.6%) | – |
| Ever used HIVST | 257 (25.2%) | – |
| Ever purchased HIVST | 159 (15.6%) | 159 (61.9%) |
| Number of times using HIVST* | | |
| Used HIVST 1–3 times | 215 (21.1%) | 215 (83.6%) |
| Used HIVST 4–10 times | 20 (1.9%) | 20 (7.8%) |
| Used HIVST more than 10 times | 6 (.05%) | 6 (2.3%) |
| Source of HIVST kit* | | |
| Pharmacy | 119 (11.6%) | 119 (46.3%) |
| Clinic/hospital | 59 (5.8%) | 59 (22.9%) |
| Non-profit organization | 13 (1.3%) | 13 (5.0%) |
| Community distribution | 16 (1.6%) | 16 (6.2%) |
| Online app/website | 1 (0.0%) | 1 (0.0%) |
| Friend | 18 (1.7%) | 18 (7.0%) |
| Family member | 4 (0.0%) | 4 (1.5%) |
| Sexual partner | 25 (2.4%) | 25 (9.7%) |
| Other | 1 (0.0%) | 1 (0.0%) |

*Respondents with missing data not reported

## Factors influencing uptake of HIVST: Segmentation analysis

Five unique segments were produced: Segment 1 (*"confident HIVST users"*, n = 277, 27.1%), Segment 2 (*"support seekers"*, n = 242, 23.7%), Segment 3 (*"privacy seekers"*, n = 140, 13.7%), Segment 4 (*"anxious testers"*, n = 185, 18.1%) and Segment 5 (*"unlikely HIVST users"*, n = 177, 17.3%). No significant differences were observed in county, setting, age, gender, money earned in a year, or education level across segments (S2 Appendix). Significance test results for HIV awareness and use are shown in Table 3. Significance test results are shown for participants answering "agree/strongly agree" to each statement in the questionnaire evaluating motivations to use HIVST, barriers to use HIVST, and improvements to HIVST (Table 4).

**Segment 1, "confident HIVST users" (n = 277, 27.1% of participants).** Segment 1 had highest awareness of HIVST (88.1%) and highest use of HIVST; of those who had heard of HIVST, 43.0% had used it. Segment 1 was also the most likely to use HIVST in the future (92.5%). They were most likely to have agreed that motivators included that HIVST allowed them to keep the result to themselves (93.1%), that they could use a self-test wherever they felt comfortable (92.4%), that they could easily test before sex (88.8%), and that using HIVST was more convenient than going to a clinic (87.4%). The barrier statements they were most likely to agree with were worrying about what their partner would do if they were HIV positive (56.7%), being afraid of getting a positive result (44.0%), and not knowing what to do if they got a positive result (38.9%).

**Segment 2, "support seekers" (n = 242, 23.7% of participants).** Awareness of HIVST was significantly lower than Segment 1 (72.3%, p < .001), as was HIVST use: 23.4% of those who had heard of HIVST had ever used it (p < .001). While overall likelihood to use HIVST in the future was high (80.2%), it was also significantly lower than Segment 1 (p < .001). Segment 2 were most likely to agree with the following statements about motivations to use HIVST: that they could test whenever they were worried they might have symptoms of HIV infection (92.6%),

**Table 3. HIVST awareness and use, by segment.**

|  | Total sample | Segment 1 | Segment 2 | Segment 3 | Segment 4 | Segment 5 |
|---|---|---|---|---|---|---|
|  | n = 1,021 | n = 277 | n = 242 | n = 140 | n = 185 | n = 177 |
| **Ever heard of HIVST** |  |  |  |  |  |  |
| Yes | 803 (78.6%) | 244 (88.1%) | 175 (72.3%)*** | 111 (79.3%) | 159 (85.6%) | 114 (64.4%)*** |
| Of those heard of HIVST, have ever used HIVST | 257 (32.0%) | 105 (43.0%) | 41 (23.4%)*** | 28 (25.2%)** | 60 (37.7%) | 23 (20.1%)*** |
| **Likelihood to use HIVST in the future[a]** |  |  |  |  |  |  |
| Unlikely | 140 (13.7%) | 3 (1.0%) | 21 (8.7%)* | 6 (4.3%) | 9 (4.9%) | 101 (57.1%)*** |
| Somewhat likely | 150 (14.7%) | 18 (6.5%) | 26 (10.7%) | 33 (23.6%)*** | 24 (12.9%) | 49 (27.7%)*** |
| Likely | 721 (70.6%) | 256 (92.5%) | 194 (80.2%)*** | 99 (70.7%)*** | 152 (82.2%)* | 20 (11.2%)*** |

Asterisks indicate level of significance between segments based on one-way ANOVA.

* $p < .05$

** $p < .01$

*** $p < .001$

[a] Excludes those who reported "Don't know/prefer not to answer"

**Table 4. Motivations and barriers to use HIVST, by segment.**

|  | Total sample | Segment 1 | Segment 2 | Segment 3 | Segment 4 | Segment 5 |
|---|---|---|---|---|---|---|
|  | n = 1,021 | n = 277 | n = 242 | n = 140 | n = 185 | n = 177 |
| **MOTIVATIONS TO USE HIVST** | | *% of participants who agree/strongly agree* | | | | |
| I can use a self-test wherever I feel comfortable | 887 (86.9%) | 256 (92.4%) | 209 (86.4%) | 126 (90.0%) | 171 (92.4%) | 125 (70.6%)*** |
| Using an HIV self-test kit allows me to keep the result to myself | 906 (88.7%) | 258 (93.1%) | 211 (87.2%) | 132 (94.3%) | 171 (92.4%) | 134 (75.7%)*** |
| Using an HIV self-test kit is more convenient than going to a clinic | 754 (73.8%) | 242 (87.4%) | 166 (68.6%)*** | 116 (82.8%) | 140 (75.7%)* | 90 (50.8%)*** |
| I prefer interacting with pharmacists over clinic staff and nurses | 491 (48.1%) | 170 (61.4%) | 104 (42.9%)*** | 77 (55.0%) | 78 (42.2%)*** | 62 (35.0%)*** |
| I'd like to test for HIV myself before going to test in a clinic | 650 (63.7%) | 212 (76.5%) | 138 (57.0%)*** | 102 (72.8%) | 138 (74.6%) | 60 (33.9%)*** |
| I can easily test before having sex | 824 (80.7%) | 246 (88.8%) | 215 (88.8%) | 116 (82.3%) | 155 (83.8%) | 102 (57.6%)*** |
| I can test whenever I am worried I might have symptoms of HIV infection | 836 (81.9%) | 241 (87.0%) | 224 (92.6%) | 124 (88.6%) | 156 (84.3%) | 91 (51.4%)*** |
| HIV self-test kits are easy to use | 626 (61.3%) | 212 (87.4%) | 126 (52.1%)*** | 80 (57.1%)*** | 136 (73.5%) | 72 (40.7%)*** |
| **BARRIERS TO USE HIVST** |  |  |  |  |  |  |
| I would rather test with an HIV counsellor present than by myself | 551 (53.9%) | 98 (35.4%) | 183 (75.6%)*** | 61 (43.6%) | 91 (49.2%)* | 118 (66.7%)*** |
| I don't know where to get the kits | 388 (38.0%) | 67 (24.2%) | 109 (45.0%)*** | 68 (48.6%)*** | 51 (27.6%) | 93 (52.5%)*** |
| I don't think I have seen them in pharmacies | 477 (46.7%) | 91 (32.9%) | 147 (60.7%)*** | 77 (55.0%)*** | 60 (32.4%) | 102 (57.6%)*** |
| I didn't know they existed | 237 (23.2%) | 41 (14.8%) | 66 (27.3%)** | 37 (26.4%) | 26 (14.1%) | 67 (37.8%)*** |
| I don't want to use a blood-based kit | 226 (22.1%) | 45 (16.2%) | 70 (28.9%)** | 40 (28.6%)* | 28 (15.1%) | 43 (24.3%) |
| I worry the results might be inaccurate | 445 (43.6%) | 72 (25.9%) | 151 (62.3%)*** | 58 (41.4%)* | 63 (34.1%) | 101 (57.1%)*** |
| I don't want someone in my house to find it | 429 (42.0%) | 76 (27.4%) | 96 (39.7%)* | 77 (55.0%)*** | 92 (49.7%)*** | 88 (49.7%)*** |
| I don't know what to do with sharps | 270 (26.4%) | 37 (13.4%) | 67 (27.7%)** | 53 (37.9%)*** | 44 (23.8%) | 69 (38.9%)*** |
| I don't want to be seen at the pharmacy getting one | 303 (29.7%) | 47 (16.9%) | 51 (21.1%) | 60 (42.9%)*** | 59 (31.9%)** | 86 (48.6%)*** |
| The type of kit I would like to use isn't in stock | 101 (9.9%) | 11 (3.9%) | 11 (4.5%) | 15 (10.7%) | 32 (17.3%)*** | 32 (18.1%)***‘ |
| I am afraid of getting a positive result | 589 (57.7%) | 122 (44.0%) | 149 (61.6%)*** | 111 (79.3%)*** | 99 (53.5%) | 108 (61.0%)** |
| I don't know what to do if I get a positive result | 526 (51.5%) | 108 (38.9%) | 136 (56.2%)*** | 94 (67.1%)*** | 81 (43.8%) | 107 (60.4%)*** |
| I worry about what my partner would do if I was HIV positive | 677 (66.3%) | 157 (56.7%) | 172 (71.1%)** | 114 (81.4%)*** | 122 (65.9%) | 112 (63.3%) |
| I don't want to spend money buying one | 291 (28.5%) | 42 (15.2%) | 70 (28.9%)** | 50 (35.7%)*** | 40 (21.6%) | 89 (50.3%)*** |
| I don't want to use a test kit that is so big I can't hide it | 370 (36.2%) | 72 (25.9%) | 77 (31.8%) | 66 (47.1%)*** | 70 (37.8%) | 85 (48.0%)*** |

Asterisks indicate level of significance between segments based on one-way ANOVA.

* $p < .05$

** $p < .01$

*** $p < .001$

that they could easily test before having sex (88.8%), that HIVST allows them to keep the result to themselves (87.2%) and that they could test wherever they feel comfortable (86.4%). Compared to Segment 1, Segment 2 was significantly less likely to agree that using HIVST was more convenient than going to a clinic (68.6%, p < .001), that they prefer interacting with pharmacists over clinic staff and nurses (42.9%, p < .001), that they'd like to test themselves before going to test in a clinic (57.0%, p < .001) and that HIVST was easy to use (52.1%, p < .001). Compared with Segment 1, Segment 2 reported significantly higher agreement with statements about barriers to using HIVST, including that they would rather test with an HIV counsellor present than by themselves (75.6%, p < .001), worrying about what their partner would do if they were HIV positive (71.1%, p = .0048), worrying the results might be inaccurate (60.7%, p < .001) and thinking they had not seen them in pharmacies (60.7%, p < .001).

**Segment 3, "privacy seekers" (n = 140, 13.7% of participants).** Awareness of HIVST was like Segment 1 (79.3%), but use was significantly lower: of those who had heard of HIVST, only 25.2% had used it (p = .007). Segment 3 was significantly less likely to use HIVST in the future: 23.6% were somewhat likely to use it (p < .001) and 70.7% were likely to use it (p < .001). Like Segment 1, they were most likely to agree that HIVST allowed them to keep the result to themselves (94.3%), that they could use a self-test wherever they felt comfortable (90.0%), that they could easily test before sex (82.3%), and that using HIVST was more convenient than going to a clinic (82.8%). However, they were significantly less likely to agree that HIVST was easy to use (57.1%, p < .001). A moderate proportion of Segment 3 agreed with statements about barriers to HIVST. They were most likely to agree with the statement that they worried about what their partner would do if they were HIV positive; this was highest of all segments (81.4%, p < .001). Relative to other segments, they were more concerned about privacy: they were most likely to agree that they did not want someone in their house to find a kit (55.0%, p < .001), that they did not want to be seen at the pharmacy getting one (42.9%, p < .001), and more likely to not want to use a kit so big they could not hide it (47.1%, p < .001).

**Segment 4, "anxious testers" (n = 185, 18.1% of participants).** Awareness of HIVST was high and like Segment 1 (85.6%), as was usage of HIVST (37.7% of those who had heard of HIVST). However, while future likelihood of use was high overall (82.2%), it was significantly lower than Segment 1 (p = .027). Like Segments 1 and 3, a high proportion of Segment 4 agreed with statements about motivations to use HIVST. They were most likely to agree that they could use a self-test wherever they felt comfortable (92.4%), that using HIVST allowed them to keep the result to themselves (92.4%) and that they could easily test before having sex (83.8%). They were significantly less likely to agree that using HIVST was more convenient than going to a clinic (75.7%, p < .036) and to prefer interacting with pharmacists over clinic staff and nurses (42.2%, p < .001). They were less likely than Segment 1 to agree that that HIVST was easy to use (73.5%) it was not significantly lower. Like other segments, they were most likely to agree that they worried about what their partner would do if they were HIV-positive (65.9%) and that they were afraid of getting a positive result (53.5%). They were significantly more likely to agree that they preferred testing with an HIV counsellor present than by themselves (49.2%, p = .021), that they did not want someone in their house to find it (49.7%, p < .001), and that they did not want to be seen at the pharmacy getting a kit (31.9%, p < .001).

**Segment 5, "unlikely HIVST users" (n = 177, 17.3% of participants).** Segment 5 also had lowest awareness of HIVST (64.4%, p < .001) and lowest HIVST use (20.1%, p < .001). They were significantly more likely to report they would not use HIVST in the future (57.1%, p < .001). Segment 5 agreed significantly less with statements about motivations to use HIVST, but they were most likely to agree they could use a self-test wherever they felt

comfortable (70.6%, p < .001) and that using HIVST allowed them to keep the result to themselves (75.7%, p < .001). They least agreed with statements that they would like to test for HIV themselves before going to test for HIV in a clinic (33.9%, p < .001) and that they preferred interacting with pharmacists over clinic staff/nurses (35.0%, p < .001). A significantly higher proportion of Segment 5 agreed with statements about barriers to use, including that they would rather test with an HIV counsellor present than by themselves (66.7%, p < .001), that they were afraid of getting a positive result (61.0%, p = .0027), that they did not know what to do if they got a positive result (60.4%, p < .001), that they did not think they had seen HIVST in pharmacies (57.6%, p < .001) and that they worried the results would be inaccurate (57.1%, p < .001).

## Propensity to pay

Participants were asked whether they would definitely or probably would buy a HIVST self-test kit if it was available for sale at a variety of costs points. 89.8% of participants would definitely or probably pay 100 KSH (1 USD). 64.7% were or probably likely to pay 300 KSH (3 USD) but dropped to 46.1% at a price of 500 KSH (5 USD). Propensity to pay further decreased at 700 KSH (7 USD) (26.2%) and 900 KSH (9 USD) (18.9%) (Fig 1).

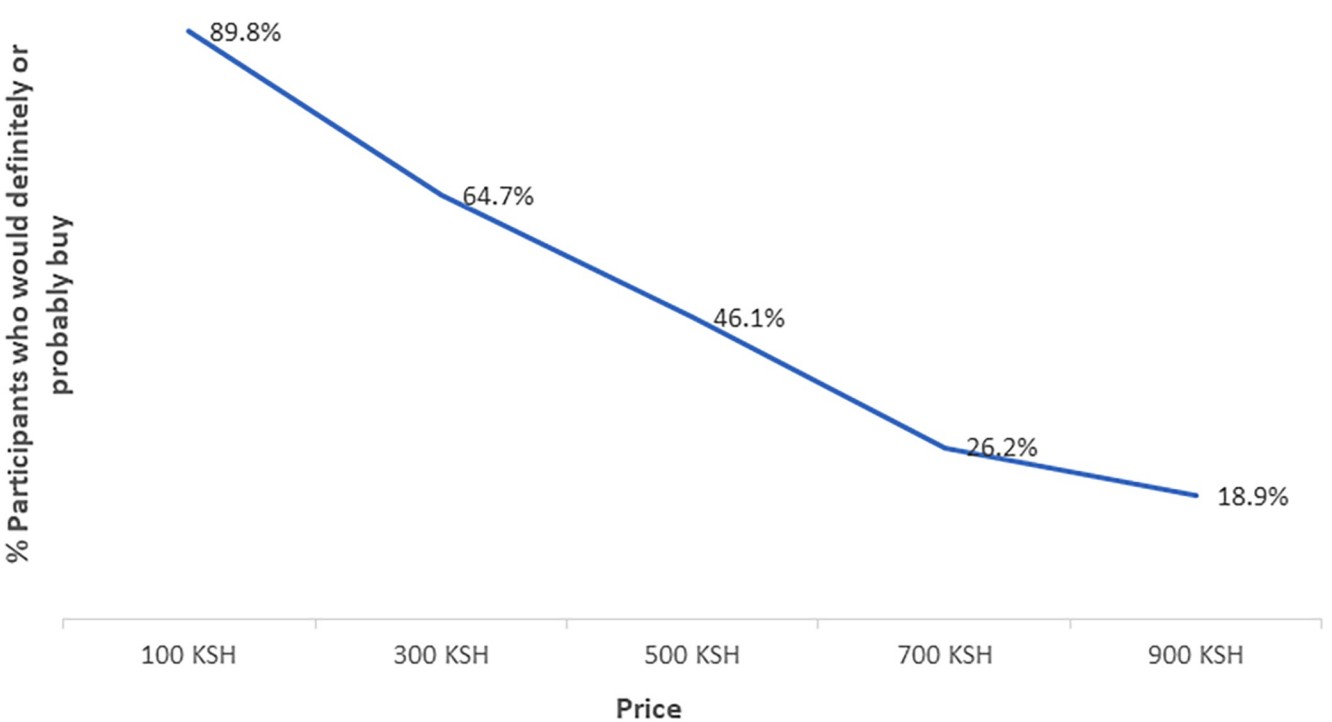

**Fig 1. Willingness to pay for HIVST by price (KSH).**

## Discussion

This research helps to characterise awareness, consideration, and usage of HIVST in Nairobi and Kisumu, as well as explore motivators and barriers to use, both attitudinal and practical. Taken together, we can understand what the current barriers to uptake of HIVST might be and which interventions would be most effective at creating increased demand for HIVST.

This research identified that awareness of HIVST among our sample in Nairobi and Kisumu was high: nearly 8 in 10 of participants had ever heard of HIVST, and confirmed that the benefits of HIVST are well understood and appreciated (e.g. ability to test in private and have control over the testing process and results [5, 30], ease of use and convenience [31, 32]). One-quarter of our sample (25.2%) had ever used HIVST; and the majority of those (83.6%) had used it 1–3 times in the last 12 months. Through the attitudinal segmentation, five unique groups of potential users were identified: Segment 1 were active HIVST users, and Segment 5 were unlikely to ever use HIVST. Segments 2, 3 and 4 may be stuck in the "consideration" phase of using HIVST due to three primary barriers: need for additional provider support, (Segment 2), need for privacy and confidentiality (Segment 3) and fear of a positive result or having to disclose a positive result (Segment 4). These three segments represent a large group of potential HIVST users should their barriers be addressed.

Segment 2's need for provider support is already well documented in the literature: other studies have found barriers to HIVST use include thinking health care providers are more knowledgeable and being afraid of misinterpreting results [31], and being afraid of testing for HIV without a counsellor present. Many patients prefer using pharmacies over health facilities due to convenience, quicker services, greater perceived privacy, and responsiveness of pharmacy personnel [33–35]. Research has also demonstrated that it is feasible to refer pharmacy clients for HIVST in Kenya [36], and that HIVST in pharmacies is highly acceptable [33]. Pharmacy-based HIVST distribution, which has already been introduced in the private sector in Kenya, is a viable option for Segment 2 users who would prefer to use HIVST in the presence of a healthcare provider but who do not need to go to the clinic to test. However, there is need to understand more from the perspective of pharmacists if this task-shifting is tenable for long-term and widespread scale-up. Alternatively, offering HIVST demonstrations in a clinical setting with potential users may alleviate some anxiety about using test kits in the future.

Segment 3, who were more likely to want privacy and confidentiality while testing, was notably also interested in ability to buy HIVST kits online, which has been shown to improve adoption [13]. More visible promotion of Kenyan online pharmacy apps (such as MYDAWA) that allow users to buy HIVST kits discreetly may help increase testing uptake within this group. Other considerations to maintain their privacy, such as nondescript or neutral packaging, may also alleviate some anxiety about confidentiality of being seen with a test kit.

It is important to note that one barrier to HIVST that was common across segments, but especially high among Segment 4, was a fear of testing positive for HIV and worry about their partner's reaction if they tested positive. Previous research has shown that fear and anxiety of a positive result, and potential social and psychological harms due to testing positive, have been documented as barriers to HIVST in other research as well [31, 32]. Making clear the linkage and support options available to HIVST users will be critical to promoting testing uptake, as well as emphasising that HIVST is a screening test and confirmatory testing in a clinic is required after a positive screen. Other studies have noted that for HIVST to translate to engagement with HIV services beyond distribution of test kits, systems must provide support with testing processes such as phone-based support and virtual/local supervision [37]. Telephone hotlines and WhatsApp chatbots have been established to support HIVST users in Kenya but may again need additional promotion and visibility so that potential consumers are aware of these services, thus encouraging

them to buy and use kits. However, additional interventions to reduce stigma around HIV diagnoses are clearly needed in tandem with these solutions. Building on U = U [38] may go a long way in allaying fears of HIV infection as individuals will understand that availability of ART and adherence can ensure people with HIV can live long, healthy lives.

While the WHO recently announced it will make an HIVST available for US $1 to the public sector [39], private sector prices remain a barrier to uptake. A 2018 willingness-to-pay study conducted in Kenya which found that median willingness to pay for HIVST was 100 KSH (1 USD) [26]. In this research, while many participants would definitely or probably pay between 100–300 KSH (1–3 USD), less than half would pay 500 KSH (5 USD), and only 19% would pay 900 KSH (9 USD). A current scan of HIVST prices on the MYDAWA app show that kits range from 470–670 KSH (4.7–6.7 USD), well above the price threshold our participants were likely to pay and limiting the ability of HIVST kits to be sold at volume. It is important to note that for the 34.0% of our sample who make up no more than 30,000 KSH per year, one test kit at the price of 500 KSH represents nearly 2% of their annual income, making frequent purchasing of HIVST unrealistic.

One final consideration that may warrant further research is how risk perception impacts attitudes toward HIVST, and willingness to pay for HIVST. This is particularly important in the context of Kenya and other countries that achieved significant gains in their HIV response, where young people may feel HIV is an issue of the past. Encouraging people to spend money on testing for something that doesn't feel like an imminent risk may be a significant challenge among the general population.

The limitations of this research should be addressed. Any self-report survey may be subject to potential biases, particularly social desirability bias, especially when discussing highly personal matters such as sexuality and sexual behaviour. To combat this, we used highly trained interviewers with experience conducting research on sensitive topics, including sexual health. Another limitation is that we did not collect data on number and nature of refusals, so we cannot comment on the nature of people opting out of this research and how it may have influenced our results We used targeted quota sampling to achieve a sample proportionate to the number of people living in Nairobi and Kisumu, and did not weight responses, therefore the larger number of respondents from Nairobi may skew the results. The segmentation analysis process produced several segment solutions and the final one was chosen by the research team based on ease of understanding and translation into public health practice [18, 20], which may have introduced bias into the results. Lastly, we excluded PrEP users from this sample as our assumption was that they would be more likely to access clinic-based HIV testing as part of routine PrEP provision; however, they may be an important group of HIVST users in the long-term as they can employ HIVST for self-care in between visits for PrEP. Additional research should examine the interest of PrEP users in utilising HIVST in Kenya.

## Conclusion

HIVST is a critical tool to increase HIV testing uptake and achieve the 95-95-95 goals. Awareness of HIVST in our sample was high, and 24% reported ever using HIVST. However, there are still many undecided potential HIVST users. Using an attitudinal segmentation, three unique barriers to HIVST uptake were identified among potential users (need for HCP support, need for increased privacy and confidentiality, and concern around a positive result/disclosure). However, the price of HIVST kits will remain a barrier for potential users. Price reduction or subsidization coupled with interventions to decrease these identified barriers and raise awareness of services available could increase HIVST uptake.

## Supporting information

**S1 Data. Data set.**
(SAV)

**S1 Appendix. Target and achieved sample.**
(DOCX)

**S2 Appendix. Demographics by segment.**
(DOCX)

## Acknowledgments

The authors thank Dr Hildah Essendi and Ronald Ngetich for their support in project set-up and management, and for Neil Martin at Ipsos for his support with study design. The authors are grateful to the interviewers and participants for their participation in this research.

## Author Contributions

**Conceptualization:** Emma Goldwin, Heather Awsumb, Sunny Sharma.

**Data curation:** Harmon Momanyi.

**Formal analysis:** Rebecca L. West, Lily Freeman, Joanna Ciecielag, Mary-Clare Ridge.

**Funding acquisition:** Heather Awsumb, Sunny Sharma.

**Investigation:** Rebecca L. West, Charlotte Pahe, Heather Awsumb.

**Methodology:** Rebecca L. West, Lily Freeman, Charlotte Pahe, Harmon Momanyi, Catherine Kidiga, Joanna Ciecielag, Heather Awsumb, Sunny Sharma.

**Project administration:** Rebecca L. West, Harmon Momanyi, Catherine Kidiga.

**Supervision:** Catherine Kidiga, Sunny Sharma.

**Visualization:** Rebecca L. West.

**Writing – original draft:** Rebecca L. West.

**Writing – review & editing:** Rebecca L. West, Lily Freeman, Charlotte Pahe, Harmon Momanyi, Catherine Kidiga, Serah Malaba, Mary-Clare Ridge, Emma Goldwin, Heather Awsumb, Sunny Sharma.

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
