## [Decision Letter · Decision Letter 0]

5 Feb 2023

PGPH-D-22-01720

Characterising the HIV self-testing market in Kenya: awareness and usage, barriers and motivators to uptake, and propensity to pay

Dear Dr. West,

Thank you for submitting your manuscript to PLOS Global Public Health. After careful consideration, we feel that it has merit but does not fully meet PLOS Global Public Health’s publication criteria as it currently stands. Therefore, we invite you to submit a revised version of the manuscript that addresses the points raised during the review process.

The reviewers feel that the rationale for the approach is not appropriately elaborated and that the Methods section lacks details. They suggest adding a description of the household level clustering of sampling and explaining how that was accounted for within your study. They also recommend including more details on the interview process, e.g. whether the interviews were conducted face-to-face.

We look forward to receiving your revised manuscript.

Kind regards,

Alex Schaefer, PhD

Associate Editor

Journal Requirements:

1. Please send a completed 'Competing Interests' statement, including any COIs declared by your co-authors. If you have no competing interests to declare, please state "The authors have declared that no competing interests exist". Otherwise please declare all competing interests beginning with the statement "I have read the journal's policy and the authors of this manuscript have the following competing interests:"

3. Please provide separate figure files in .tif or .eps format only and remove any figures embedded in your manuscript file. Please also ensure that all files are under our size limit of 10MB.

4. We noticed that you used "data not shown and unpublished data" in the manuscript. We do not allow these references, as the PLOS data access policy requires that all data be either published with the manuscript or made available in a publicly accessible database. Please amend the supplementary material to include the referenced data or remove the references.

5. Please amend your Data Availability Statement and indicate where the data may be found

Additional Editor Comments (if provided):

Reviewers' comments:

Reviewer's Responses to Questions

**Comments to the Author**

1. Does this manuscript meet PLOS Global Public Health’s publication criteria? Is the manuscript technically sound, and do the data support the conclusions? The manuscript must describe methodologically and ethically rigorous research with conclusions that are appropriately drawn based on the data presented.

Reviewer #1: Partly

Reviewer #2: Yes

2. Has the statistical analysis been performed appropriately and rigorously?

Reviewer #1: I don't know

Reviewer #2: Yes

3. Have the authors made all data underlying the findings in their manuscript fully available (please refer to the Data Availability Statement at the start of the manuscript PDF file)?

Reviewer #1: Yes

Reviewer #2: Yes

4. Is the manuscript presented in an intelligible fashion and written in standard English?

Reviewer #1: Yes

Reviewer #2: Yes

5. Review Comments to the Author

Reviewer #1: I think the topic is of great interest but the rationale for the approach is not clear as currently written. A few things would increase enthusiasm for this paper

- a description of household level clustering of sampling and how that was accounted for

- refusal rate for enrollment

- description of KSH to USD or some other currency for comparison

- why the segmental analysis was pursued and why this is a helpful way to look at the data

- why PrEP users were excluded

- would hvae been interested in seeting the data for the whole group on some of the varaibels collected like motivations and barriers to HIVST use,

Reviewer #2: On the whole, this is a well-designed and implemented study. Perhaps what is a little unclear is exactly how were respondents to the study recruited. It is difficult to understand where the interviewers located the respondents. Was it at their houses? How were the interviews done i.e., were they paper based or electronic capture. Was itself administered interviews or face-to-face? Were there any declines? Just some precision in the description needed.

6. PLOS authors have the option to publish the peer review history of their article (what does this mean?). If published, this will include your full peer review and any attached files.

**Do you want your identity to be public for this peer review?** For information about this choice, including consent withdrawal, please see our Privacy Policy.

Reviewer #1: No

Reviewer #2: **Yes: **Makobu Kimani

---

## [Decision Letter · Decision Letter 1]

10 Mar 2023

Characterising the HIV self-testing market in Kenya: awareness and usage, barriers and motivators to uptake, and propensity to pay

PGPH-D-22-01720R1

Dear Ms West,

We are pleased to inform you that your manuscript 'Characterising the HIV self-testing market in Kenya: awareness and usage, barriers and motivators to uptake, and propensity to pay' has been provisionally accepted for publication in PLOS Global Public Health.

Best regards,

Moses Okumu

Academic Editor

Good job with addressing the reviewers' comments. Please proofread your paper or use an editor to further strengthen its sharpness.

Reviewer Comments (if any, and for reference):

Reviewer's Responses to Questions

**Comments to the Author**

1. If the authors have adequately addressed your comments raised in a previous round of review and you feel that this manuscript is now acceptable for publication, you may indicate that here to bypass the “Comments to the Author” section, enter your conflict of interest statement in the “Confidential to Editor” section, and submit your "Accept" recommendation.

Reviewer #1: All comments have been addressed

Reviewer #2: All comments have been addressed

2. Does this manuscript meet PLOS Global Public Health’s publication criteria? Is the manuscript technically sound, and do the data support the conclusions? The manuscript must describe methodologically and ethically rigorous research with conclusions that are appropriately drawn based on the data presented.

Reviewer #1: Yes

Reviewer #2: Yes

3. Has the statistical analysis been performed appropriately and rigorously?

Reviewer #1: I don't know

Reviewer #2: Yes

4. Have the authors made all data underlying the findings in their manuscript fully available (please refer to the Data Availability Statement at the start of the manuscript PDF file)?

Reviewer #1: Yes

Reviewer #2: Yes

5. Is the manuscript presented in an intelligible fashion and written in standard English?

Reviewer #1: Yes

Reviewer #2: Yes

6. Review Comments to the Author

Reviewer #1: (No Response)

Reviewer #2: The authors have made effort to address my comments. I will defer to the second reviewer on the final decision. The authors should in future consider more meaningful involvement of local investigators. It may not quite appear appropriate to have just two local investigators as middle authors. This is in keeping with the philosophy upon which PLOS GPH was founded.

7. PLOS authors have the option to publish the peer review history of their article (what does this mean?). If published, this will include your full peer review and any attached files.

**Do you want your identity to be public for this peer review?** For information about this choice, including consent withdrawal, please see our Privacy Policy.

Reviewer #1: No

Reviewer #2: No
